# Tardive Oromandibular Dystonia Induced by Trazodone: A Clinical Case and Management from the Perspective of the Dental Specialist

**DOI:** 10.3390/toxins14100680

**Published:** 2022-09-30

**Authors:** Nicolás P. Skarmeta, Giannina C. Katzmann, Constanza Valdés, Dominique Gaedechens, Francisca C. Montini

**Affiliations:** 1Facultad de Odontología, Universidad de los Andes, Santiago 7620001, Chile; 2Hospital del Salvador, Providencia 7500922, Chile; 3OPH Clinic, Vitacura 7630000, Chile

**Keywords:** oromandibular dystonia, tardive dystonia, trazodone, botulinum toxin

## Abstract

Background: Tardive Oromandibular Dystonia is an iatrogenic drug-induced movement form of extrapyramidal symptoms associated primarily with chronic consumption of dopamine receptor blocking agents. Tardive symptoms attributable to selective serotonin reuptake inhibitors antidepressants are far less prevalent. Clinical Case: The authors will present a clinical case and management, from the dental specialist perspective, of a 55-year-old female patient who developed tardive oromandibular dystonia induced by Trazodone prescribed for sleep insomnia. Conclusions: Trazodone-induced oromandibular dystonia is extremely rare. Early identification and assessment of tardive symptoms are imperative for successful treatment. Trazodone should be prescribed with caution in patients taking other medications with the potential to cause tardive syndromes.

## 1. Introduction

Oromandibular movement disorders (OMD) are a group of hyperkinetic motor disorders which affect the muscles of mastication, the face, and the tongue. From an etiologic standpoint, these disorders may be acquired, inherited, or idiopathic [1].

Acquired OMD can be attributed to perinatal brain injuries, infections, trauma, ischemic/hemorrhagic cerebrovascular accidents, arteriovenous malformations, toxicity, and drugs [2].

Drug-induced OMD encompasses a wide variety of hyperkinetic and hypokinetic extrapyramidal involuntary movements, manifesting in the form of dystonia, tremors, stereotypy, tics, dyskinesia, or a combination of different types of abnormal movements [3]. The magnitude and complexity of these abnormal movements can vary from being almost unnoticeable to producing complete social impairment.

The onset of these iatrogenic movement disorders has been primarily associated with the consumption of dopamine receptor blocking agents (typical and atypical neuroleptics) and other medications such as antidepressants (e.g., tricyclic antidepressants, selective serotonin reuptake inhibitors, and selective norepinephrine serotonin reuptake inhibitors), antiemetics, and other medications used in gastrointestinal disorders [4].

Since the motor response to the offending drug is highly variable among patients, it is easier to categorize drug-induced OMD based on the temporal profile (i.e., the patient’s exposure to the drug). Acute drug-induced OMD often occurs within hours or days after the exposure to the offending drug; in subacute drug-induced OMD, symptoms appear more slowly within days or weeks after the exposure; and tardive OMD often refers to chronic exposure to the drug, at least three months or one month in patients older than 60 years of age [3,5].

Diagnosis and recognition of these involuntary motor disorders can be elusive. In this article, the authors will present a clinical case of tardive oromandibular dystonia attributable to Trazodone prescription for sleep insomnia.

## 2. Clinical Case

A 55-year-old female patient was referred to the orofacial pain unit of the OPH clinic for showing symptoms of temporomandibular joint arthralgia and osteoarthritis, orofacial myofascial pain, and sleep bruxism. An experienced Orofacial Pain Specialist performed a comprehensive clinical history and examination following the Diagnostic Criteria for Temporomandibular Disorders (DC/TMD) protocol and the International Classification of Orofacial Pain-1 (ICOP-1) criteria. At the moment of the examination, the patient was taking 150 mg of Venlafaxine for major depressive disorder, 4 mg of Clonazepam for sleep onset insomnia, and hormone replacement therapy for menopause.

Another major complaint of the patient was difficulty initiating and maintaining sleep. Thus, an Epworth Sleepiness Scale and Insomnia Severity Index were performed to complete the sleep evaluation. Due to the high scores presented by the patient and clinical history consistent with chronic sleep insomnia, polysomnography (PSG) was requested Even though the PSG revealed no sleep breathing disorders, the exam showed severe modifications in the patient’s sleep architecture, a mild increase in the arousal index, and low-frequency sleep bruxism. (Table 1) The PSG results led us to hypothesize that the patient’s sleep architecture alterations were because of the chronic consumption of benzodiazepines. Thus, the patient was derived to her treating psychiatrist to gradually diminish and eliminate the intake of Clonazepam.

During this time, the patient successfully underwent her osteoarthritis and orofacial myofascial pain treatment. In addition, an upper oral occlusal splint was prescribed to manage low-frequency sleep bruxism.

In order to manage sleep initiation and maintenance insomnia, the psychiatrist tapered the dosage of Clonazepam and progressively replaced it with Trazodone until a target dose of 100 mg was reached. Throughout this period, the patient did not experience any extrapyramidal symptoms.

Almost three months later, after the end of the orofacial pain treatment, the patient returned complaining of a sudden worsening of her awake bruxism. The patient presented abrupt rhythmic involuntary muscle contractions, similar to jaw-closing and jaw-deviating dystonic-like movements affecting the right side of her masticatory muscles. (See Figure 1, and Appendix A) The dystonic-like episodes were spontaneous or triggered by everyday activities such as talking, chewing, and yawning. The severity of these motor contractions tended to be exacerbated by emotional stress and nervousness produced by these uncontrollable movements. During these episodes, some typical features of oromandibular dystonia were noticed, such as some sensory tricks (gentle pressure on the inferior incisors would milden the jaw-deviating dystonia as the placement of a tongue depressor between the molars for jaw-closing dystonia), and phenomena (e.g., oromandibular dystonia overflowing the right eyelid, resulting in right blepharospasm).

A video examination protocol suggested by Yoshida was carried out to assess the type and magnitude of involuntary movements [6]. In addition, a complete haematological and biochemical blood examination was required and all the parameters were within normal range. Thus, both thyroid and parathyroid hormonal dysfunction were ruled out.

Since the literature indicates that Trazodone can induce extrapyramidal symptoms and abnormal oromandibular movement disorders only start after the addition of this medication. The authors, in agreement with her treating physician, suggested performing a slow taper until complete discontinuation of Trazodone was achieved.

A progressive dose reduction of 25 mg of Trazodone was scheduled every two weeks. In the meantime, an inferior occlusal splint was prescribed, considering that mild contact in the inferior incisor tended to ease the intensity of the dystonic-like movements.

During the progressive discontinuation of Trazodone, the authors noticed an immediate attenuation of the dystonic symptoms. However, only achieving a partial but clear remission of them. No video protocol was recorded during the tampering of Trazodone.

Considering that the patient’s mental health status was stable, it was decided to take a more conservative approach, only eliminating the most likely offending drug, namely, Trazodone. Consequently, the authors maintained that Venlafaxine intake was unchanged in order not to jeopardize the psychiatric treatment, and also closely monitored the evolution of the clinical symptoms. Additionally, the authors indicated the administration of 25 units of incobotulium toxin (Xeomin^®^, Merz, Germany) per masseter muscle, 20 units per temporalis muscle, and 15 units were applied to the inferior head of the right lateral pterygoid with electrical stimulation guidance (See anatomical landmarks in Figure 2).

During the two-week follow-up consultation, the authors could recognize a substantial decrease in the magnitude and intensity of the oromandibular movement. Two months and seven months after the discontinuation of the offending drug and botulinum toxin application, the authors re-recorded Yoshida’s video protocol to carry out an objective before and after comparison of the treatment results. See Table 2 for comparison prior to and after the treatment.

## 3. Discussion

Antidepressant Drug-induced extrapyramidal symptoms are considered a rare cluster of severe adverse reactions which cause considerable morbidity and significantly impair the quality of life [7].

Tardive syndromes are usually produced by chronic exposure to dopamine receptor blocking agents (DRBA), such as typical and atypical antipsychotics. However, other medications such as tricyclic antidepressants (TCAs), serotonin reuptake inhibitors antidepressants (SSRIs), and medications used for gastrointestinal disorders can also produce extrapyramidal symptoms [3,4].

The plausible mechanisms involved in iatrogenic drug-induced movements disorders are usually related to a drug-induced extended occupancy of dopamine D2 receptors, which may be related to a hypothetical post-synaptic supersensitivity and upregulation of dopamine receptors, resulting in the disinhibition of the globulus pallidum internus and the subthalamic nucleus, which, in turn, can produce a wide variety of hyperkinetic manifestations [3].

In particular, the theoretical link between SSRIs and the onset of tardive syndromes may be related to the antagonism of 5HT2 receptors in the GABA-ergic interneurons of the nigrostriatal pathway, which can lead to the suppression of the activity of these dopaminergic neurons explaining the occurrence of hyperkinetic movement disorders. The incidence of extrapyramidal movement disorders with these pharmacological compounds is relatively low. In fact, the estimated risk of developing these disorders is lower than 1 in 1000 [8].

Trazodone is a triazolopyridine derivative from the serotonin 5HT2A and 5HT2c receptor antagonist and belongs to the reuptake inhibitors class of antidepressants, approved for major depressive disorder. Additionally, it is commonly used as an off-label medication to treat delirium and insomnia, because it displays a moderately potent antagonist effect over alpha 1A and 2A adrenoreceptors and H1 histaminergic receptors, which is probably related to its sedative effect [9,10].

Another plausible mechanism that may explain how Trazodone may trigger extrapyramidal symptoms is its inhibitory effect on T-type calcium channels, exerting a similar effect in the subthalamic nucleus similar to other medications related to tardive syndromes such as haloperidol and flunarizine [11,12].

To the authors’ knowledge, there is scarce literature reporting trazodone-induced tardive symptoms. Furthermore, the authors only found a few reports of tardive dystonic symptoms after chronic consumption of Trazadone [12,13].

Treatment recommendations in tardive syndromes often start with progressive tampering with the offending drug. In fact, the sooner the causative drug is removed, the likelier the tardive symptoms will be solved.

Pharmacological management often varies depending on the severity of the symptoms, the type of drug inducing the symptoms (i.e., DRBAs, TCAs, SSRIs), and the chronicity of intake.

The most common therapeutic approach suggested in the literature, besides a progressive taper of the offending drug, is the use of Dopamine-depleting agents such as Tetrabenazine and Reserpine; Antiparkinson agents such as Amantadine; GABA agonists such as Clonazepam, Valproic Acid, and Baclofen; Anticholinergic agents such as Trihexyphenidyl, and Chemodenervation with botulinum toxin [3].

In this clinical case, the oromandibular movement disorders were consistent with tardive dystonia. Consequently, the authors chose the abovementioned treatment strategy based on the phenomenology of tardive symptoms. In this clinical case, we performed chemodenervation using botulinum toxin in the dystonic masticatory muscles [14], installed a diurnal inferior oral appliance (to produce a sensory trick and protect the oral structures) [15], and carried out a progressive reduction and removal of the offending drug. The treatment successfully achieved a complete attenuation of the abnormal movement disorders.

## 4. Conclusions

Tardive syndromes may severely affect the quality of life and produce substantial social impairment. Hence, treating tardive symptoms is complex, especially because sometimes symptoms may be treated successfully. Thus, preventing and identifying the onset of tardive syndromes is of paramount importance. Although the risk of producing tardive syndrome induced by Trazodone is extremely low, clinicians should balance the risk/benefit ratio before prescribing drugs that can potentially produce tardive symptoms. Moreover, prescribing Trazodone for insomnia or delirium should be avoided completely or done only when it is strictly necessary for patients with the chronic intake of other drugs with the potential to induce tardive syndromes. The authors’ recommendation to health professionals, based on this unusual clinical case, is to avoid drug-on-drug interactions whenever possible and prefer medications that have less potential of producing tardive syndromes.

## Figures and Tables

**Figure 1 toxins-14-00680-f001:**
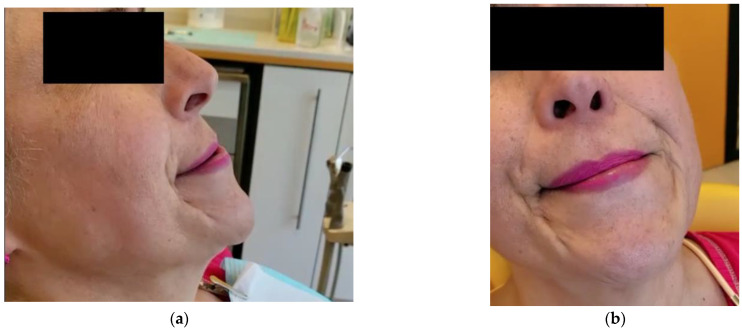
(**a**) Jaw-deviating and closing dystonia producing episodes of forceful clothing and protrusive movements; (**b**) Jaw-deviating dystonia to the left side producing forceful contractions of the inferior head of the right lateral pterygoid muscle.

**Figure 2 toxins-14-00680-f002:**
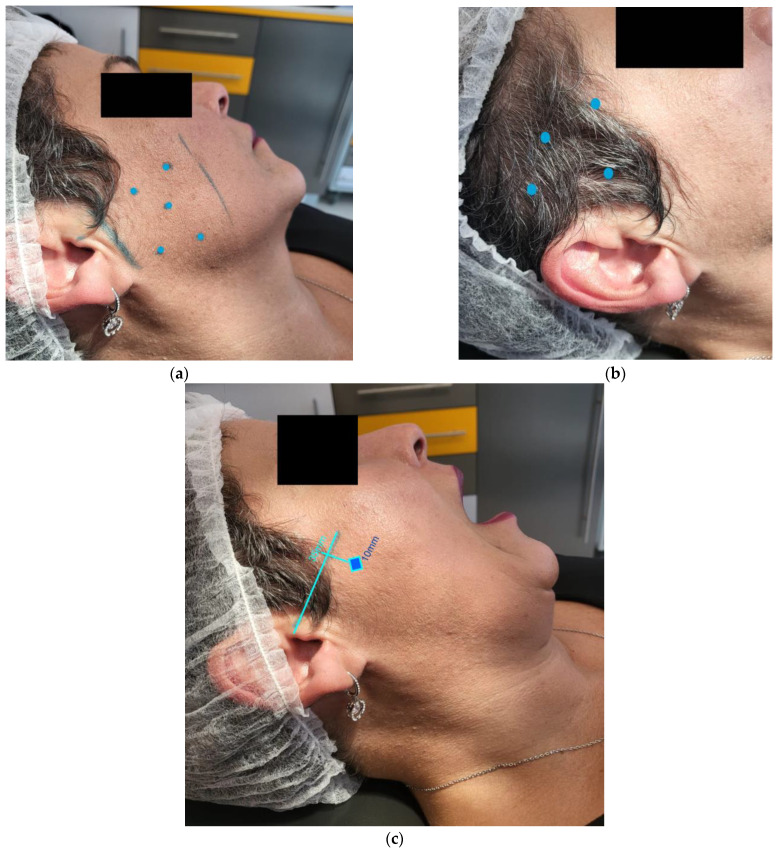
(**a**) Incobotulinum toxin injection sites for the masseter muscle; (**b**) Incobotulinum toxin injection sites for the temporalis muscle. (**c**) Anatomic landmarks for extraoral injection with electrical stimulation guidance to the inferior head of the right lateral pterygoid muscle.

**Table 1 toxins-14-00680-t001:** REM sleep stages, Arousal Index, Sleep Bruxism Index.

Sleep Stages	Arousal Index	Sleep Bruxism Index
NREM1 (10.0%)	16.4 eps per hour	2.4 eps per hour
NREM2 (75.9%) NREM3 (14.3%) REM (0.0%)		

**Table 2 toxins-14-00680-t002:** Yoshida’s Video examination Protocol for OMD.

Examination Protocol for OMD	Prior Treatment *	Two-Month Follow-Up *	Seven-Month Follow-Up *
At rest (10 s)	4	0	0
Counting 1 to 10 out loud	2	0	0
Open/Close mouth (5 times)	2	0	0
Lateral Movements (5 times)	3	0	0
Jaw Protrusion (5 times)	5	0	0
Tongue Protrusion (hold for 5 s)	0	0	0
Hold a long vowel (5–10 s)	0	0	0
Read a Sentence	2	0	0
Gum chewing (30 s)	4	0	0

* Number of involuntary dystonic episodes triggered by different mandibular activities.

## Data Availability

Not applicable.

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
