# Peer review of "Tardive Oromandibular Dystonia Induced by Trazodone: A Clinical Case and Management from the Perspective of the Dental Specialist"

_toxins, 2022, doi:10.3390/toxins14100680_

Round 1
Reviewer 1 Report
The case is interesting. However, I have several comments:
1) it must be taken in consideration that the patient had previously bruxism and was intaking venlafaxine. In a PubMed search, there are 14 references regarding induction of bruxism by venlafaxine. For this reason, the possibility that the aggravation of bruxism/oromandibular dystonia should be related with the interaction between trazodone and venlafaxine.
2) The follow-up period of 2 months, considering that the patient received infiltrations with incobotulinum toxin (the effect of the infiltration could have persisted), is not enough to conclude that the patient improved by trazodone withdrawal. The authors should specify what happened in the next months
3) Consider to add a series on Drug-induced tardive syndromes (PMID: 18591121) in the reference list
Author Response
We want to thank the reviewer for analyzing our manuscript.
1) Indeed, intake of Venlafaxine is associated with the induction of bruxism and the incidence of extrapyramidal symptoms. We will clarify in the corrected manuscript certain aspects of how the clinical symptoms evolved to address the reviewer's comments.
Also, we know that there is literature reporting that both Trazodone and Velenfaxine can produce extrapyramidal symptoms. Moreover, we agree with the reviewer that synergistic interaction between these two medications can occur. We think that queries are addressed in the corrected manuscript.
The reduction of Venlafaxine was unnecessary since the dystonic symptoms started to diminish after the progressive dose tampering of Trazadone.
The patient did have definitive low-frequency sleep bruxism. We decided, in agreement with the treating psychiatrist, not to tamper the dose of Velenfaxine, considering that the Major depressive disorder was under control with this medication.
2) We added a seven-month control.
3) We added the PUBMED series to the reference list. (PMID: 18591121)
Reviewer 2 Report
I would like to congratulate the authors on this case report. It is a very interesting case, very well written and pleasant to read.
I would have a major concern and some additional comments
- what is current follow up? If only 2 months one cannot be certain if the patient improved due to botulin toxin effect (may last until 6 months in some patients) or due to trazodone interruption. If the follow up is of only 2 months the authors cannot assume the OMD was due to trazodone, as there are alternative etiologies. To be considered. It would be needed at least 6 months of follow-up (preferably more) to be sure it was due to trazodone.
- in the first video there’s the impression of excessive eye blinking, was this the case? If so, one could consider the diagnosis of Meige syndrome. A full video of the patient, including the eyes (if she consents) would be must important
- what was the time interval between the onset of trazodone and the OMD?
- was a cerebral brain MRI or CT scan performed? It would be of upmost importance to rule out structural lesions
Author Response
Reviewer 2
We want to thank the reviewer for the good comments.
- We address minor language errors and typos.
- We address the concerns about the follow-up period, making another video recording in a 7-month control. We think this is enough time to suggest that the intake of Trazodone was related to the induction of OMD.
- The reviewer is correct; the dystonic movements also affected the eyelid as part of the overflown phenomena. We added this information to the corrected manuscript. The patient did not permit us to upload videos showing her entire face. The patient approved the videos uploaded in the supplemental files but did not waive her permission for new videos.
- We agree with the reviewer that neuroimaging can be crucial in discarding structural lesions (tumors, bleeding, strokes, atrophic lesions, neurodegeneration, or accumulation of metals such as copper or iron. Especially, a Mri could be very useful in discarding neurodegenerative disorders that could produce secondary OMD. However, we would like to stress that the most prevalent form of secondary dystonia is, in fact, tardive dystonia, accounting for 35% of the cases, followed by perinatal injuries or a history of encephalitis (30%), followed by stroke (13%); Other forms of secondary OMD a far less prevalent, such as Wilson disease (less than 6 %). Also, not all neural lesions result in dystonia (basal ganglia, cerebellum, brainstem lesions, frontal and parietal lobe lesions); depending on location, these lesions are more related to more extensive forms of dystonia. Heredodegenerative disorders which can produce dystonia as a feature of more syndromic manifestations. In that line, some clinical clues may indicate secondary dystonia secondary to structural lesions such as rapid progression or progression to generalized, prominent oro-bulbar involvement, other neurological signs (peripheral neuropathy, deafness, parkinsonism, supranuclear gaze palsy, among others), and adult-onset of leg dystonia. During the treatment, we did not discard the possibility of soliciting neuroimages; however, since none of these symptoms were present and we had a good response with the elimination of the Trazodone, and during the follow-up, we decided not to solicit complementary neuroimaging examination.
Round 2
Reviewer 1 Report
The authors addressed correctly my concerns. However, I was not able to see the inclusion of reference PMID: 18591121 among the citations as was stated.
Reviewer 2 Report
I would like to acknowledge the authors effort on replying to all the reviewers concerns. Still, from an educational point of view it is my opinion an MRI or at least CT scan has to be performed to exclude basal ganglia lesions.